# Comparative Proteomics of *Marinobacter* sp. TT1 Reveals Corexit Impacts on Hydrocarbon Metabolism, Chemotactic Motility, and Biofilm Formation

**DOI:** 10.3390/microorganisms9010003

**Published:** 2020-12-22

**Authors:** Saskia Rughöft, Nico Jehmlich, Tony Gutierrez, Sara Kleindienst

**Affiliations:** 1Microbial Ecology, Center for Applied Geosciences, University of Tübingen, 72076 Tübingen, Germany; saskia.rughoeft@uni-tuebingen.de; 2Department of Molecular Systems Biology, Helmholtz Centre for Environmental Research–UFZ GmbH, 04318 Leipzig, Germany; nico.jehmlich@ufz.de; 3Institute of Mechanical, Process & Energy Engineering, Heriot-Watt University, Edinburgh EH14 4AS, UK; tony.gutierrez@hw.ac.uk

**Keywords:** *Marinobacter*, dispersant, Corexit, WAF, hexadecane, hydrocarbon metabolism, proteomics, chemotactic motility, biofilm formation

## Abstract

The application of chemical dispersants during marine oil spills can affect the community composition and activity of marine microorganisms. Several studies have indicated that certain marine hydrocarbon-degrading bacteria, such as *Marinobacter* spp., can be inhibited by chemical dispersants, resulting in lower abundances and/or reduced biodegradation rates. However, a major knowledge gap exists regarding the mechanisms underlying these physiological effects. Here, we performed comparative proteomics of the Deepwater Horizon isolate *Marinobacter* sp. TT1 grown under different conditions. Strain TT1 received different carbon sources (pyruvate vs. *n*-hexadecane) with and without added dispersant (Corexit EC9500A). Additional treatments contained crude oil in the form of a water-accommodated fraction (WAF) or chemically-enhanced WAF (CEWAF; with Corexit). For the first time, we identified the proteins associated with alkane metabolism and alginate biosynthesis in strain TT1, report on its potential for aromatic hydrocarbon biodegradation and present a protein-based proposed metabolism of Corexit components as carbon substrates. Our findings revealed that Corexit exposure affects hydrocarbon metabolism, chemotactic motility, biofilm formation, and induces solvent tolerance mechanisms, like efflux pumps, in strain TT1. This study provides novel insights into dispersant impacts on microbial hydrocarbon degraders that should be taken into consideration for future oil spill response actions.

## 1. Introduction

Chemical dispersants are routinely applied during major marine oil spills to break up surface slicks and disperse oil in the water column. Following the Deepwater Horizon (DWH) oil spill in the Gulf of Mexico in 2010, for instance, seven million liters of dispersants (Corexit EC9500A and EC9527A) were applied in response to the release of an estimated 800 million liters of crude oil into the Gulf ecosystem [1,2]. Dispersant exposure has consistently been shown to alter marine microbial community dynamics and to select for specific taxa responding to inputs of petroleum hydrocarbons (HCs) [3,4,5,6,7,8]. However, studies have produced conflicting results on how dispersants might affect the HC biodegradation potential of microbial communities, with findings ranging from enhanced [9,10] to unaffected [8,11] or decreased HC degradation activities [4], and the underlying mechanisms remaining unresolved.

Dispersant impacts have also been highlighted by several studies that demonstrated inhibition effects of dispersants on certain species/strains of HC-degrading bacteria, observed in both pure cultures [12,13,14] and seawater microcosm experiments [3,7]. Members of the genus *Marinobacter*, in particular, have been shown to become negatively affected by chemical dispersant exposure [3,4,7,13,15]. This genus includes several ubiquitous marine HC degraders that often represent alkane-degrading key players responding to oil spillage in the marine environment [4,8,16]. Recent work in our group with *Marinobacter* sp. strain TT1 has furthermore demonstrated that this strain is able to grow on Corexit EC9500A as sole carbon and energy source [17], similar to other marine HC-degrading isolates belonging to the genera *Colwellia*, *Alcanivorax*, or *Acinetobacter* that were shown to utilize components of the dispersant mixture [14,18].

The metabolic pathways of HC biodegradation are relatively well characterized in several HC-degrading isolates (reviewed in Reference [19,20]). Aerobic alkane biodegradation, for example, is typically performed via a sequential oxidation process by a few key enzymes (i.e., alkane monooxygenases or cytochrome P450 oxidases, alcohol dehydrogenases and aldehyde dehydrogenases) and connects to the cytosolic fatty acid metabolism. However, it remains largely unknown how chemical dispersants, such as Corexit, might affect the metabolism and cellular processes of HC degraders and, in turn, how exposure leads to the different observed microbial responses. While the exact composition of chemical dispersants remains proprietary, Corexit EC9500A was reported to contain a petroleum distillate fraction, propylene glycols, and different anionic and nonionic surfactants [21,22]. These components may themselves cause detrimental effects to microbial cells even in the absence of crude oil, since surfactants are known to induce several potentially cytotoxic effects by partitioning within cell membranes, potentially impairing permeability processes and the functioning of membrane proteins [23,24,25]. Solvent stress caused by high concentrations of (individual or a mixture of) HCs is known to trigger specific adaptations or avoidance responses in HC-degrading bacteria, such as efflux pump expression or negative chemotactic behavior [26,27,28,29], as well as changes in membrane lipid composition [30]. Furthermore, chemically dispersed oil can exhibit synergistic toxicity to marine microorganisms, i.e., higher toxicity than oil alone [31,32,33].

The alkane degrader *Marinobacter* sp. TT1 was isolated during the DWH oil spill [34] and recent work in our group has shown that its growth and biodegradation of *n*-hexadecane can be detrimentally impacted by Corexit exposure [17], which may explain why this strain did not become enriched during the DWH spill [34]. Since the genomic potential of strain TT1 and the underlying mechanisms of Corexit affecting its physiology and metabolism remain unknown, we conducted a comparative proteomics study in order to, for the first time, (i) characterize the HC-degrading metabolism of *Marinobacter* sp. TT1 based on proteomics, (ii) identify which of the previously reported components of Corexit might be metabolized by this strain, and (iii) elucidate potential effects of Corexit exposure on the strain’s cellular processes. For this, the protein profiles of *Marinobacter* sp. TT1 were analyzed when grown in the presence or absence of Corexit EC9500A (Cxt), on pyruvate or *n*-hexadecane (Hxdc), or using a crude oil water-accommodated fraction (WAF) or chemically enhanced WAF (CEWAF; containing Corexit EC9500A) that simulate crude oil and/or Corexit exposure in the marine water column in the event of an oil spill at sea.

## 2. Materials and Methods

### 2.1. Bacterial Strain and Culture Conditions 

The strain *Marinobacter* sp. TT1 (=*Marinobacter* sp. DSM 26291) was isolated from deep-sea plume water samples collected during the Deepwater Horizon oil spill, using *n*-hexadecane as enrichment substrate [34]. It is closely related to *M. salarius* R9SW1 and *M. algicola* DG893 with 99.43% and 99.07% 16S rRNA gene sequence identity, respectively. In this study, strain TT1 was cultivated on ONR7a minimal medium [35], supplemented with different carbon substrates, and grown (dark, 20 °C, 120 rpm on shaker) in half-filled 20 mL glass headspace vials (baked at 300 °C for 8 h) with PTFE-lined crimp lids. For inoculation, 200 μL of the respective pre-cultures were transferred. 

For the proteomics experiment, four different pre-cultures (originating from the same glycerol stock) were used. They were adapted to grow on one of the following carbon substrates, respectively: pyruvate, *n*-hexadecane, Corexit EC9500A, or crude oil water-accommodated fraction. At the start of the experiment, all pre-cultures were three days old and had been grown without prior Corexit exposure, with the exception of the Corexit-adapted pre-culture. Dutch Underground Consortium (DUC) crude oil was used to prepare the WAF solution and the CEWAF solution (additionally containing Corexit EC9500A) for the experiment as described in Reference [4]. 

### 2.2. Experimental Design 

At the start of the experiment, the following treatments for *Marinobacter* sp. TT1 were established in half-filled 20 mL glass headspace vials (Appendix A) using four different pre-cultures, adapted to different carbon substrates. The *n*-hexadecane-adapted pre-culture was used to inoculate (1) cultures with no added carbon substrate (Control), (2) cultures with 100 mg L^−1^
*n*-hexadecane (Hxdc), and (3) cultures with 100 mg L^−1^
*n*-hexadecane and 10 mg L^−1^ Corexit (Hxdc+Cxt). The Corexit-adapted pre-culture was used to establish treatments with (4) 100 mg L^−1^ Corexit (Corexit), whereas the WAF-adapted pre-culture was used to inoculate treatments with either (5) 1 mL of WAF solution (6 mg L^−1^ WAF-derived dissolved organic carbon) or (6) 200 µL of CEWAF solution (6 mg L^−1^ CEWAF-derived dissolved organic carbon). The pyruvate-adapted pre-culture was used to inoculate cultures containing (7) 3 mM (264 mg L^−1^) pyruvate. All treatments were run in triplicate and sampled sacrificially after 0 and 4 days, except for the pyruvate-containing treatments which were sampled after 1 day due to a faster growth with this substrate, as determined in pre-experiments. Due to concerns about gas-phase losses during sampling, separate triplicate cultures were prepared for treatments (2) and (3), in order to separately obtain samples for cell counts and HC quantification analysis. Separate replicate cultures were prepared for protein analysis from all seven treatments and each sampled after 1 day (pyruvate treatments) or 4 days (all other treatments). Additionally, abiotic controls were prepared for HC or pyruvate quantification, containing no inoculum and either (8) 100 mg L^−1^
*n*-hexadecane (ab. Hxdc), (9) 100 mg L^−1^
*n*-hexadecane and 10 mg L^−1^ Corexit (ab. Hxdc+Cxt), or (10) 3 mM of pyruvate.

### 2.3. Cell Counts 

For cell counts, samples were fixed with 1% paraformaldehyde (PFA) and stored at 4 °C until further processing. To reduce cell aggregates, 1% (*w*/*v*) ethylenediaminetetraacetic acid (EDTA) was added to the samples before sonication (20% intensity, 30 s; Sonoplus ultrasonic homogenizer, Bandelin electronic GmbH & Co. KG, Berlin, Germany), a procedure that was optimized for this culture. Samples were then filtered onto Isopore polycarbonate membrane filters (GTTP type, 0.2 μm; Merck Millipore, Darmstadt, Germany) and stained with 4′,6-diamidino-2-phenylindole (DAPI; 1 μg mL^−1^) for 10 min, washed with ddH_2_O for 5 min, rinsed in ethanol (80%), and air dried in the dark at room temperature. Membrane filters were embedded using a 1:4 mixture of Vectashield mounting medium (Vector Laboratories, Burlingame, CA, USA) and Citifluor AF2 glycerol solution (EMS Acquisition Corp., Hatfield, PA, USA) before the slides were analyzed using fluorescence microscopy (Leica DM 5500 B; Leica Microsystems, Wetzlar, Germany). Images were taken at a magnification of 1000x with a Leica DFC 360 FX camera using the Leica Application Suite Advanced Fluorescence software (2.6.0.766). Cell counts of the images were performed using the ‘Find Maxima’ function (noise tolerance = 7) of the Fiji distribution of ImageJ [36], counting a minimum of 20 images and 600 cells per sample.

### 2.4. Quantification of Pyruvate and Hydrocarbon Concentrations

Concentrations of pyruvate were quantified using high-performance liquid chromatography (HPLC; Shimadzu Prominence, Kyoto, Japan) equipped with an Aminex HPX 87H column (Bio Rad, Vienna, Austria) and an SPD-M10A VP photo-diode array detector (flow rate 0.6 mL min^−1^; oven at 40 °C, 5 mM H_2_SO_4_ as eluent).

For *n*-hexadecane quantification, deuterated *n*-hexadecane (D34, Sigma-Aldrich, St. Louis, MO, USA) was added to the samples as internal standard (20 mg L^−1^) before extracting the entire vial using 9 mL cyclohexane (purity 99.9%, Carl ROTH, Karlsruhe, Germany). Vials were shaken at 300 rpm for half an hour, phases were allowed to separate overnight, and subsamples of the cyclohexane were used to quantify the remaining *n*-hexadecane concentrations via gas chromatography (GC; Agilent 6890N GC) coupled with mass spectrometry (MS; Agilent 5973 MS). For separation, a J + W Scientific DB-5MS (30 m length, 0.25 mm ID, 0.25 µm film thickness) capillary column was used. The device was operated in a pulsed splitless mode with a Helium flow of 0.8 mL/min. Oven temperature was initiated at 65 °C (4 min), then ramped at 10 °C/min to 220 °C, further ramped at 20 °C/min to 310 °C, and held at this temperature for 5 min. 

### 2.5. Protein Extraction and Proteome Analysis

For protein analysis, 100 mL of culture replicates were pooled per sample triplicate, filtered onto Sterivex filters (Merck Millipore, Darmstadt, Germany) and immediately frozen. The filters were cut into small pieces and dissolved in 1 mL lysis buffer (8 M Urea, 2 M Thiourea, 1 mM phenylmethylsulfonyl fluoride, PMSF). Cells on the filters were disrupted by bead beating (FastPrep-24, MP Biomedicals, Santa Ana, CA, USA; 5.5 ms, 1 min, 3 cycles), followed by ultra-sonication (UP50H, Hielscher, Teltow, Germany; cycle 0.5, amplitude 60%) and centrifugation (10,000× *g*, 10 min). The protein lysate was loaded on SDS-gel and run for 10 min. The gel piece was cut, washed, and incubated with 25 mM 1,4-dithiothreitol (in 20 mM ammonium bicarbonate) for 1 h and 100 mM iodoacetamide (in 20 mM ammonium bicarbonate) for 30 min, and destained, dehydrated, and proteolytically cleaved overnight at 37 °C with trypsin (Promega, Walldorf, Germany). The digested peptides were extracted and desalted using ZipTip-μC18 tips (Merck Millipore, Darmstadt, Germany). The peptide lysates were re-suspended in 0.1% formic acid and analyzed by nanoliquid chromatography mass spectrometry (LC-MS/MS; UltiMate 3000 RSLCnano, Dionex, Thermo Fisher Scientific, Waltham, MA, USA). Mass spectrometric analyses of eluted peptide lysates were performed on a Q Exactive HF mass spectrometer (Thermo Fisher Scientific, Waltham, MA, USA) coupled with a TriVersa NanoMate (Advion, Ltd., Harlow, UK). LC gradient, ionization mode and mass spectrometry mode were used as described in Reference [37]. The mass spectrometry proteomics data have been deposited to the ProteomeXchange Consortium via the PRIDE [38] partner repository with the dataset identifier PXD021108.

### 2.6. Data Analysis

Data resulting from LC-MS/MS measurements were analyzed with the Proteome Discoverer (v.2.4, Thermo Fischer Scientific) using SEQUEST HT. The protein-coding sequences of the reference UniProt proteome of *Marinobacter* sp. DSM 26291 (= strain TT1; protein-coding sequence entries 4,109, TaxID: 1761792) were used as database. Search settings were set to trypsin (Full), max. missed cleavage: 2, precursor mass tolerance: 10 ppm, fragment mass tolerance: 0.02 Da. The obtained label-free quantification intensities were further analyzed using Perseus v1.6.1.5 [39]. For statistical tests, abundance data were log2 transformed, normalized (median-centered) and filtered. Only proteins with at least five peptide spectrum matches (PSM) and identified in at least two biological triplicates of at least one growth condition were considered for statistical analysis. A permutation-based false discovery rate (FDR) approach was used to identify proteins with significantly different abundances between growth conditions, while correcting for multiple testing (parameters: S0 = 0, FDR 0.05, 1,000 randomizations). First, a multiple sample test (ANOVA) was performed on all 17 samples (one outlier had to be removed) grouped by treatment to gain an overview of the data. Hierarchical clustering was performed using the Spearman correlation–based distance (300 clusters, 10 iterations). Subsequently, two-sample tests (Student’s *t*-test) were run for specific pairwise sample comparisons of interest for this study’s objectives, followed by Tukey’s honestly significant difference post hoc tests (*q* < 0.05). Proteins of interest were further characterized using the blastp suite [40], the UniParc clustering feature of UniProt [41], KEGG’s KOALA (KEGG Orthohology And Links Annotation [42]) and pathway mapper [43] tools. The genetic organization of gene clusters of interest was assessed using the genome of *Marinobacter* sp. DSM 26291 (= strain TT1; IMG genome ID: 2619618959) and the JGI IMG/M database [44]. The R programming language [45] was used to produce all presented data plots. 

## 3. Results

### 3.1. Growth and Substrate Utilization of Marinobacter sp. TT1 

Growth of *Marinobacter* sp. TT1 was observed on all substrates, reaching similar orders of magnitude for almost all growth conditions. The treatments supplied with *n*-hexadecane ± Corexit, Corexit only, or pyruvate had reached cell numbers of 10^8^ cells ml^−1^ at the proteome sampling timepoints (1.40–3.75 × 10^8^ cells ml^−1^; Figure 1A). Different macroscopic growth patterns were observed between these treatments with a classical homogeneous increase in optical density in pyruvate-containing cultures and growth in differently sized aggregates in the other three treatments (Corexit: barely visible aggregates; *n*-hexadecane: thin biofilm at water-HC interface and approximately 1-mm-sized aggregates in medium; *n*-hexadecane + Corexit: large, thin aggregates ranging from 1–1.5 cm size; Appendix A). Even though cell numbers were lower in WAF and CEWAF treatments, growth was still observed after four days when samples were taken for proteomics (from 6.00 × 10^5^ cells ml^−1^ to 4.72 or 5.10 × 10^6^ cells ml^−1^, respectively). Biodegradation of pyruvate (61% after one day; Figure 1B) or *n*-hexadecane (46% or 67% degraded after four days, respectively, in treatments with or without Corexit; Figure 1B) was confirmed via GC-MS or HPLC measurements for the respective treatments. 

### 3.2. Overview of Proteomic Analysis

The proteomic analysis resulted in a total of 3,008 proteins (ranging from 1 to 4289 peptide-spectrum matches per protein), representing a recovery of 73% of *Marinobacter* sp. TT1′s proteome (4109 proteins; UniProtKB proteome UP000199211) (Appendix A). Distinct protein profiles were detected for all treatments, while similar proteome patterns were observed in biological replicate samples (Appendix A). Across the different treatments (i.e., Pyruvate, Hxdc, Hxdc+Cxt, Corexit, WAF, and CEWAF), 2154 proteins were significantly differentially abundant according to a multiple sample test (ANOVA; permutation-based FDR < 0.05). However, due to the much lower biomass available in WAF and CEWAF samples, and the higher inter-replicate variability, proteome data from WAF and CEWAF treatments was mainly compared to each other and interpreted with caution. When considering specific pairwise sample comparisons (Student’s *t*-test, permutation-based FDR < 0.05), a total of 1140 proteins were detected in significantly different abundances during growth on *n*-hexadecane compared to the non-HC control (pyruvate) with 50% of them in higher abundance on *n*-hexadecane (Appendix A). When considering the effects of Corexit exposure, 36 proteins were significantly differentially abundant during growth on *n*-hexadecane with Corexit compared to the *n*-hexadecane treatments (with 55% of them more abundant with Corexit; Appendix A). Additionally, 1286 proteins were significantly differentially abundant during growth on Corexit compared to growth on *n*-hexadecane with 43% more abundant on Corexit (Appendix A). Only two proteins were detected as significantly differentially abundant between the WAF and CEWAF treatments (Appendix A). 

### 3.3. Proteome of Marinobacter sp. TT1 Grown on n-Hexadecane

The protein profile of *Marinobacter* sp. TT1 grown on *n*-hexadecane indicated upregulated metabolisms for alkane degradation and alginate biosynthesis, with additional upregulated processes including peptidoglycan and lipopolysaccharide (LPS) synthesis, as well as oxidative stress responses (Figure 2 and Appendix A). Only one of the alkane 1-monooxygenases encoded in the genome of *Marinobacter* sp. TT1 (UniProt ID: A0A1I4KVH1) was more abundant during growth on *n*-hexadecane than in the non-HC control treatment (log_2_ fold change [FC] = 3). This alkane 1-monooxygenase was found to have 83% amino-acid sequence identity to the AlkB2 enzyme (A0A455W7K7) of *Marinobacter hydrocarbonoclasticus* YB03 and might also be homologous to AlkB2 (Q0VTH3) of *Alcanivorax borkumensis* SK2 (59.03% amino-acid sequence identity). Additionally, a putative P450 cytochrome alkane hydroxylase (A0A1I4KEY9) was detected in significantly higher abundance compared to the non-HC control (FC = 4.5). Two proteins putatively involved in the transport of alkanes into the cell were also significantly more abundant during growth of strain TT1 on *n*-hexadecane: a long-chain fatty acid transport protein (FC = 4.9; A0A1I4JKC5) and an Ig-like domain containing protein (FC = 4.6; A0A1I4JKB4). These were identified as putative homologues of the alkane uptake proteins AupA (H8WEC1; 83% amino-acid sequence identity) and AupB (H8WEC0; 56% amino-acid sequence identity), respectively, which were previously described in *Marinobacter hydrocarbonoclasticus* SP17 [46]. The electron transfer proteins rubredoxin-NAD+ reductase (A0A1I4K5F9), ferredoxin (A0A1I4MH37), and ferredoxin-NADP reductase (A0A1I4KTN3) were likewise significantly more abundant during growth on *n*-hexadecane (FC ≥ 1.2) and thus likely involved in the first terminal *n*-hexadecane oxidation step. Finally, four alcohol dehydrogenases (A0A1I4J3P7, A0A1I4ISL3, A0A1I4MI03, A0A1I4H5P2) and one aldehyde dehydrogenase (A0A1I4K5K4), probably involved in the subsequent oxidation steps of *n*-hexadecane metabolism, were also significantly more abundant (FC = 0.7 to 5.0). Interestingly, a large number of proteins associated with the extracellular polysaccharide (EPS) alginate involved in biofilm formation were significantly more abundant during growth on *n*-hexadecane compared to pyruvate (FC = 2.1 to 5.8). This included (homologues of) the following proteins involved in the biosynthesis and export of the polysaccharide alginate: AlgD (A0A1I4LFN3), Alg44 (A0A1I4LF83), AlgE (A0A1I4LF78), AlgF (A0A1I4LFF1), AlgA (A0A1I4JXV5, A0A1I4LGW8), AlgC (A0A1I4LG98), and the regulatory proteins MucB (A0A1I4IUP7), AlgR (A0A1I4JGY9), and AlgZ (A0A1I4JH21). These proteins were found to be encoded by three different alginate-related gene clusters in the genome of *Marinobacter* sp. TT1 (*algD844KEGIJFXLAC* [SAMN04487868_11444-56], *algU-mucABC* [SAMN04487868_104201-04], and *algRZ* [SAMN04487868_10662-63]) with the first gene cluster showing a notably similar organization (Appendix A) to the alginate operons described in *Pseudomonas aeruginosa* [47] and *Alcanivorax borkumensis* [48]. All abundant proteins from this first putative operon (AlgD, Alg44, AlgE, AlgF, AlgA, AlgC) were also detected in significantly higher abundances during growth on *n*-hexadecane when compared to the Corexit-only treatment (FC = 1.9 to 6.3). Furthermore, a number of proteins required for peptidoglycan or lipopolysaccharide (LPS) biosynthesis, five different phospholipases and several enzymes possibly involved in mitigating oxidative stress, were found to be significantly more abundant in the *n*-hexadecane treatment compared to the pyruvate treatment (FC = 0.4 to 3.7; Appendix A).

### 3.4. Protein Profiles of Marinobacter sp. TT1 during Corexit Exposure

The protein profile of *Marinobacter* sp. TT1, when influenced by dispersant exposure, pointed towards upregulated alkane and non-linear HC metabolism; and different transporter systems, as well as pronounced chemotaxis and motility processes (Figure 2). 

#### 3.4.1. Corexit versus n-Hexadecane

The alkane 1-monooxygenase (A0A1I4KVH1) and putative P450 cytochrome alkane hydroxylase (A0A1I4KEY9) that were abundant during growth on *n*-hexadecane were significantly less abundant in cultures grown on Corexit (FC = 4.3 and 1.2, respectively). However, the second alkane 1-monooxygenase (A0A1I4KFD0) was significantly more abundant in the Corexit-only treatment (FC = 8). This enzyme has only low amino-acid sequence identity (36.07%) to the other alkane 1-monooxygenase of *Marinobacter* sp. TT1 (A0A1I4KVH1) but is 99.75% identical (amino-acid sequence identity) to the AlkB1 enzyme of *Marinobacter hydrocarbonoclasticus* VT8 (A1TXS2) and is a homologue of AlkB1 (Q0VKZ3) from *Alcanivorax borkumensis* SK2 (87.16% amino-acid sequence identity). A number of proteins encoded by the genes located downstream from the AlkB1 homologue in the genome of strain TT1 were significantly more abundant in Corexit treatments, as well: an aldehyde dehydrogenase (FC = 4.6; A0A1I4KFQ9), a choline dehydrogenase (FC = 4.7; A0A1I4KF71), a fatty-acyl-CoA synthase (FC = 2.2; A0A1I4KER1), and an outer membrane protein with putative homology to AlkL (FC = 4.5; A0A1I4KF83). While two alcohol dehydrogenases (FC = 4.1, A0A1I4KKP8; FC = 0.5, A0A1I4H5P2) and another aldehyde dehydrogenase (FC = 1.8; A0A1I4KKB9) were also significantly more abundant, most of the proteins likely involved in *n*-hexadecane metabolism (i.e., detected in high abundances in the *n*-hexadecane-amended treatments) were less abundant in Corexit treatments. However, several enzymes potentially involved in the biodegradation and metabolism of aromatic HCs and other HC-related compounds were significantly more abundant in Corexit treatments (average FC = 1.9), including enzymes assigned to the metabolism of haloalkanes, cycloaliphatic HCs, phenols, benzoates, and aminobenzoates (Appendix A). Interestingly, a few proteins involved in sulfur metabolism were also significantly more abundant in Corexit treatments (average FC = 1.8), i.e., a sulfotransferase (A0A1I4KTW4), the sulfur carrier protein FdhD (A0A1I4NAC8), and a thioester reductase (A0A1I4JTH2). 

According to KEGG’s ortholog annotation tool (KOALA), the largest fractions of annotated proteins significantly more abundant in Corexit compared to *n*-hexadecane treatments belonged to the KEGG orthology (KO) categories of ‘signaling and cellular processes’ (24%) and ‘environmental information processing’ (22%; Appendix A). Within these categories, a large group of proteins with significantly higher abundances compared to *n*-hexadecane treatments (FC = 0.4 to 5.9) were assigned to transport systems, including 13 tripartite ATP-independent periplasmic (TRAP) type mannitol/chloroaromatic transporter proteins, 5 efflux pump proteins, and 27 amino acid ATP-binding cassette (ABC) transporter proteins (13 of these specifically for branched-chain amino acids). The second largest group (FC = 0.2 to 5.3) was assigned to cellular processes of chemotaxis and motility (i.e., 11 methyl-accepting chemotaxis proteins (MCPs), 12 type IV pilus proteins, 3 twitching motility proteins, and 5 flagellar proteins), with several of these proteins also significantly more abundant when compared to the pyruvate treatments. 

#### 3.4.2. n-Hexadecane/Corexit versus n-Hexadecane

The annotated proteins detected in significantly higher abundances in the treatment with *n*-hexadecane and Corexit compared to the *n*-hexadecane-only treatment belonged mainly to the KO categories of ‘signaling and cellular processes’ (27%) and ‘environmental information processing’ (27%; Appendix A). This included five transporter-affiliated proteins (FC = 0.5 to 2.6; A0A1I4MNH9, A0A1I4KD04, A0A1I4IF13, A0A1I4I3J1, A0A1I4H9T9) and one MCP (FC = 0.5; A0A1I4MSJ0). Interestingly, the putative P450 cytochrome alkane hydroxylase (FC = 0.7; A0A1I4KEY9) and the ferredoxin-NADP reductase (FC = 2.5; A0A1I4KTN3), abundant in *n*-hexadecane treatments, were significantly less abundant in the cultures with added Corexit. 

#### 3.4.3. CEWAF versus WAF

Even though only two proteins with significantly different abundances were detected between the more environmentally relevant treatments WAF and CEWAF, probably due to the high detected inter-replicate variance, some notable systematic differences could still be observed between treatments. Among those proteins found in higher abundance in the WAF treatment, according to a *p*-value-truncated *t*-test (62 proteins; *p* < 0.05; Appendix A), the largest annotated fractions belonged to the KO categories of ‘xenobiotics biodegradation and metabolism’ (20%) and ‘carbohydrate metabolism’ (18%; Appendix A). This included several enzymes with predicted functions in the metabolism of non-linear HCs and related compounds (Appendix A), i.e., the biodegradation of cycloaliphatics, phenols, naphthalenes, benzoates, and xylenes (average FC = 4.3). On the other hand, the proteins detected in higher abundance in CEWAF cultures (85 proteins; *p* < 0.05; Appendix A) mainly belonged to the KO categories ‘environmental information processing’ (20%) and ‘signaling and cellular processes’ (16%; Appendix A). Notable proteins from these categories included, for instance, three MCPs (FC = 0.5 to 2.1; A0A1I4LXM2, A0A1I4HAX6, A0A1I4MFB0), three flagellar proteins (FC = 0.8 to 1.4; FliG, FliK, FlgL; A0A1I4HL94, A0A1I4HME7, A0A1I4J7M9), the pilus assembly protein FimV (FC = 0.8; A0A1I4M1I2), and five proteins associated with ABC or TRAP-type transporter systems (FC = 0.9 to 2.2; A0A1I4MSA8, A0A1I4L200, A0A1I4HXQ3, A0A1I4KD04, A0A1I4IE06). Generally, many of the detected proteins associated with chemotaxis and motility were most abundant in the Corexit and CEWAF treatments when comparing all hydrocarbon-amended treatments (Figure 2). Most of the previously described proteins involved in the metabolism of *n*-alkanes were also detected in both WAF and CEWAF cultures with only one notable difference between these treatments. While both alkane 1-monooxygenases were detected in the CEWAF treatment, only the AlkB1 homologue (A0A1I4KFD0) was also detected in the WAF treatment. 

## 4. Discussion

The obtained protein profiles of *Marinobacter* sp. TT1 illustrated the strain’s ability to metabolize a wide range of HC compounds, produce alginate, utilize Corexit components as substrates, and elucidated additional physiological adaptation processes to Corexit exposure (see Figure 3).

### 4.1. Hydrocarbon Metabolism of Marinobacter sp. TT1

The *n*-hexadecane metabolism of *Marinobacter* sp. TT1 appears relatively analogous to the alkane metabolism of other marine alkane degraders [49,50,51], with our results suggesting important roles for the AupAB, AlkB2, and cytochrome P450 homologues and a number of involved dehydrogenases. The AupA homologue likely enables strain TT1 to take up *n*-hexadecane from the outer membrane, with the AupB homologue guiding the alkane to the inner membrane, where the terminal oxidation machinery is located [46]. Both the AlkB2 homologue and the putative cytochrome P450 hydroxylase of strain TT1 can likely perform the first oxidation step by utilizing either rubredoxin or ferredoxin (and the respective reductases) as redox partners to yield *n*-hexadecanol. The AlkB1 homologue might not have been detected in higher abundances in *n*-hexadecane treatments because it preferentially degrades shorter alkanes. This would mirror reports on *A. borkumensis* SK2, in which AlkB1 metabolizes shorter alkanes (C_5_-C_12_), while AlkB2 degrades longer alkanes (C_8_–C_16_) [52]. Similar patterns of monooxygenase induction by different substrate ranges have also been reported for other alkane degraders with multiple monooxygenases and hydroxylases [53,54,55]. Finally, several different dehydrogenases were detected that could perform the subsequent oxidation steps from alcohol via aldehyde to fatty acid during *n*-hexadecane biodegradation in *Marinobacter* sp. TT1. In addition, several proteins implicated in cell envelope modification, oxidative stress response, and a number of chaperones were significantly abundant during growth on *n*-hexadecane, pointing towards an involvement of these proteins in the alkane-degrading lifestyle and associated biofilm formation in bacteria like strain TT1, as previously discussed elsewhere [56,57,58,59,60].

WAFs are often used to simulate the subsurface state of crude oil in the water column in the event of an oil spill at sea. They typically contain alkanes and mostly low-molecular weight aromatic HCs, such as BTEX (benzene, toluene, ethylbenzene, xylene) or PAH (polycyclic aromatic hydrocarbons) compounds [61,62]. The proteins we detected from strain TT1 corroborate this, since enzymes assigned to the metabolism of benzoates, xylenes, naphthalene, phenols, and cycloaliphatic HCs were found in highest abundances in the WAF treatments. Only the AlkB1 homologue was detected, which likely degrades shorter alkanes (e.g., C_5_–C_12_), like the AlkB1 of *A. borkumensis* SK2 [52]. This makes sense since longer-chain alkanes are less likely to be found in WAFs due to their poor solubility. We therefore hypothesize that alkane metabolism by strain TT1 played a minor role in WAF treatments compared to the degradation of aromatics. Our findings suggest that the HC-degrading metabolic spectrum of strain TT1 is much wider than previously reported [34] and add to the growing body of evidence that members of the genus *Marinobacter* are, in addition to degrading alkanes, often also capable of utilizing aromatic HCs as sources of carbon and energy [16,63,64]. 

### 4.2. Alginate Biosynthesis of Marinobacter sp. TT1

Growth of *Marinobacter* sp. TT1 on *n*-hexadecane was apparently associated with an upregulation in alginate metabolism. Alginate is a polysaccharide composed of multiple monomer subunits which are synthesized from fructose-6-phosphate, polymerized, and exported across the outer membrane by proteins of the alginate machinery of which several homologues were detected in this study (Appendix A, reviewed in Reference [65]). Alginate is commercially extracted from seaweed for industrial applications and, thus, of biotechnological interest [66]; however, there are currently only two well-characterized bacterial producers of alginate (*Pseudomonas aeruginosa* and *Azotobacter vinelandii*). To our knowledge, this is the first evidence of an alginate operon in *Marinobacter*, as well as the induction of alginate proteins by alkane biodegradation. Evidence for a potential link between HC biodegradation and alginate gene transcription was previously very sparse [67,68]. However, alginate plays a well-documented role in the formation of resistant mucoid biofilms by pathogenic *P. aeruginosa* [69,70], and *alg*-gene mutants of *A. borkumensis* were reported to have a reduced capacity for binding to lipophilic stains [71]. Growth of *M. hydrocarbonoclasticus* SP17 on *n*-hexadecane was also shown to depend on biofilm formation for accessing non-dissolved *n*-hexadecane [49,60]. Thus, alginate likely plays an important role in the *n*-hexadecane-degrading lifestyle of *Marinobacter* sp. TT1 and probably constitutes a large part of the aggregates and biofilms observed in this study when grown on *n*-hexadecane. 

### 4.3. Biodegradation of Corexit Components by Marinobacter sp. TT1

Proteins potentially involved in the metabolism of alkanes, aromatic HCs, and surfactants were detected in the Corexit treatments, suggesting that these proteins might be involved in the utilization of dispersant components as carbon sources by *Marinobacter* sp. TT1 (Figure 3B).

#### 4.3.1. Alkane Metabolism

A number of proteins associated with the utilization of *n*-alkanes as carbon and energy sources (e.g., AlkB1 homologue, AupA/B homologues, putative P450 hydroxylase) were detected in high abundances when strain TT1 was supplied with the chemical dispersant Corexit as sole carbon substrate. This suggests that alkanes are a biodegradable part of the petroleum distillate fraction in Corexit, which was also suspected previously [21,72]. The high abundance of the AlkB1 homologue (compared to the highly abundant AlkB2 homologue in *n*-hexadecane treatments) further suggests that mainly alkanes of shorter chain-length than *n*-hexadecane were available, which is in agreement with previous reports of C_9_-C_16_ hydrocarbons making up the petroleum distillate fraction of Corexit [73].

Five proteins encoded adjacently to the AlkB1 homologue in the genome of strain TT1 had assigned functions in alkane and fatty acid metabolism, and four of them were detected in highest abundances in Corexit treatments. This may suggest that this respective gene cluster forms part of strain TT1′s *alk* operon, which would putatively be induced by alkanes shorter than *n*-hexadecane (Appendix A). This operon also included a homologue of *alkL* from *P. putida* GPo1, which encodes an outer membrane protein known for facilitating the uptake of C_7_-C_16_ alkanes [74,75]. Therefore, both the AupAB and AlkL homologues were likely involved in uptake of alkanes from Corexit across the outer membrane. The alkane metabolism detected in Corexit treatments additionally differed from the observed *n*-hexadecane metabolism of strain TT1 regarding the abundance profiles of detected alcohol and aldehyde dehydrogenases, while two other *n*-hexadecane-metabolizing enzymes were also significantly less abundant in *n*-hexadecane treatments with added Corexit. These observations might again be due to different substrate ranges of these enzymes or they could be related to other Corexit components affecting them in an unknown manner. A similar pattern of differential HC degradation gene expression was reported recently in a metatranscriptomic study, assessing dispersant impacts on marine microbial communities affected by crude oil or diluted bitumen input [8]. Thus, Corexit exposure seemingly not only supplies additional alkane substrates for marine HC degraders but may also influence their active HC metabolism in additional ways. 

#### 4.3.2. Metabolism of Other Corexit Components

A number of proteins putatively assigned to the degradation of other HCs and related compounds were detected during growth on Corexit, specifically haloalkanes, cyclohexanones, phenols, benzoates/xylenes, and N-containing aromatic HCs (i.e., nitrotoluenes, aminobenzoates), which might represent additional components of the Corexit mixture that were metabolized by strain TT1. It is furthermore possible that the surfactant constituents of Corexit, i.e., dioctyl sulfosuccinate (DOSS), Span 80, Tween 80 and Tween 85 [22], were degraded by strain TT1, either completely or partially, thus serving as additional sources of carbon for the strain. Many HC-degrading bacteria (including the closely related *Marinobacter algicola* and *M*. *salarius*) have been shown to utilize Tween surfactants during routine strain characterization [76,77]. Microbial degradation of DOSS has also been reported [7,18,78] and presumably proceeds via ester bond hydrolysis and metabolism of resulting alkyl chains and sulfosuccinates [78,79,80]. Thus, the surfactant alkyl and acyl side chains might have been degraded by the detected AlkB1 homologue, cytochrome P450 hydroxylase, or lipases, while the proteins related to sulfur metabolism detected in Corexit treatments might have been involved in DOSS metabolism. Moreover, the solvent fraction of Corexit reportedly contains propylene glycol and dipropylene glycol monobutyl ether [81], which could also have been degraded by strain TT1 as shown for other isolates [18]. In order to establish a better connection of these potential substrates to the observed proteomic response of strain TT1, further experimental studies using only specific surfactants or glycols as carbon substrates are required. In general, these novel insights into the specific catabolism of strain TT1 growing on Corexit as a sole carbon source support previous reports of enriched suspected Corexit-degraders in marine microbial communities exposed to Corexit [4,7,72]. 

### 4.4. Additional Cellular Processes Affected by Corexit Exposure in Marinobacter sp. TT1

All protein profiles from treatments with Corexit showed similar trends, suggesting that dispersant exposure affected the interaction of *Marinobacter* sp. TT1 cells with their environment, as evidenced by upregulated efflux pumps, other transmembrane transporter systems, chemotactic motility, and changed biofilm formation behavior (Figure 3B). Similar trends were also observed in two recent metatranscriptomic studies on Corexit affecting seawater microbial communities [8,82] and in situ analysis from the DWH subsurface plume [83], although it remains unclear to what extent the plume metatranscriptomic data were influenced by Corexit and/or oil exposure. 

#### 4.4.1. Transmembrane Transporter Systems

Efflux pumps play an important role in conferring solvent tolerance to bacteria, like *P. putida* [26,29], and, thus, they are probably used by *Marinobacter* sp. TT1 to expel harmful solvents or metabolic intermediates from the cell when exposed to higher concentrations of Corexit. The highly abundant TRAP-type transporters in Corexit treatments, on the other hand, likely played a role in the uptake of Corexit components since they are known to typically transport organic acids and sulfonates [84,85]. Some TRAP-type transporters were also shown to import aromatic compounds, like 4-chlorobenzoate, and lignin-derived monomers [86,87]. The majority of highly abundant transporters in Corexit treatments were amino acid/amide ABC transporters, with about half of them specifically branched-chain amino acid (BCAA) transporters. A similar enrichment of ABC transporter genes and/or transcripts was previously reported for seawater microcosms amended with Corexit and oil or bitumen [8], petroleum-contaminated microbial mats [88], dibutyl phthalate-contaminated black soils [89], and in *Sphingomonas* sp. GY2B during Tween 80-enhanced phenanthrene degradation [90]. Generally, ABC transporters accept a wide range of substrates [91,92]. It remains unclear why specifically BCAA transporters showed such high abundances in Corexit treatments, but we offer a few hypothetical scenarios: First, an increased demand for BCAAs could have been caused by increased requirements for branched-chain fatty acids which were recently linked to both the maintenance of membrane fluidity during exposure to stressors, like aromatic HCs [93,94], and a novel quorum sensing system in gram-negative bacteria [95] that affects EPS and biofilm formation behaviors [96]. Alternatively, ABC transporters can also facilitate solute efflux [97], and several microbial ABC transporters that had initially been annotated differently have been implicated in the transport of aromatic HCs, as demonstrated by experimental evidence [98,99,100]. Thus, the detected ABC transporters might have transported branched/aromatic HCs or surfactants, either as substrates into the cell or out of the cytoplasm, reinforcing the activity of efflux pumps. 

#### 4.4.2. Chemotactic Motility and Biofilm Formation

Chemotactic behavior plays an important role in HC biodegradation (reviewed in Reference [101]), by either increasing bioavailability of HC substrates (i.e., moving towards attractants) or enabling bacteria to avoid toxic HCs or high concentration of HCs (i.e., movement away from repellents). Accordingly, a few chemotaxis proteins and a number of flagellar proteins were significantly more abundant in *n*-hexadecane treatments compared to the non-HC controls. However, Corexit-containing treatments systematically showed significantly higher abundances of chemotactic motility-related proteins compared to other HC-containing treatments (i.e., *n*-hexadecane, WAF). Therefore, chemotaxis and motility were presumably involved in the response of strain TT1 to Corexit exposure. One the one hand, strain TT1 was likely chemotactically attracted to Corexit components that it could metabolize. On the other hand, a pronounced negative chemotactic response to other harmful components could explain the higher abundance of different chemotactic sensor and flagellum-regulator proteins (i.e., MCPs and CheA/Y) and would align with previous observations of Corexit inhibiting some *Marinobacter* spp. [3,4,7,13,15]. The more abundant type IV pilus and twitching motility proteins might additionally enable strain TT1 to regulate its motility in a more sophisticated manner for this purpose, similar to *P. aeruginosa* that can synergistically utilize both flagella- and pili-mediated motility mechanisms and switch between them if needed [102]. Finally, chemotaxis and motility also play important roles in biofilm establishment and formation [103,104,105].

Several of our findings point towards Corexit exposure inducing changes in cellular processes associated with aggregate/biofilm formation in strain TT1 (i.e., alginate production, chemotactic motility, putative quorum sensing processes), which likely led to the distinct observed aggregate morphologies. Hydrophobic alginate-based biofilms were probably most beneficial for growth on *n*-hexadecane [49,58,60]. In treatments with added Corexit, however, chemotactic processes might have resulted in stronger autoaggregation behavior and altered EPS production, providing a higher stress tolerance to cells utilizing *n*-hexadecane during Corexit exposure. When growing only on Corexit, HC substrates were more bioavailable and increased cell motility might have been more advantageous than aggregating when adapting to the higher Corexit concentration. This reflects the different biofilm phenotypes observed in *P. aeruginosa*, i.e., alginate-based mucoid biofilms versus type IV pili-mediated biofilms based on other exopolysaccharides [69,103]. 

### 4.5. Environmental Implications of Proteomic Findings

The DWH disaster was a significant offshore oil spill in deep waters and represented a unique opportunity to study the response of indigenous bacterial communities to a major influx of crude oil and chemical dispersants. However, many knowledge gaps remain regarding how to link the genetic capability of enriched taxa to their role in the degradation of oil and Corexit, and how to explain the observed physiological effects caused by Corexit exposure. This study represents the first proteome-level investigation addressing these questions. Our findings point towards Corexit exposure inducing changes in cellular processes, specifically those associated with aggregate/biofilm formation in strain TT1, which reinforces previous reports that exposure to Corexit, and to other synthetic chemical dispersants, can affect EPS secretion and aggregate formation by marine microorganisms [3,4,5,17,106]. These effects became highly relevant during the DWH oil spill, where Corexit strongly influenced the formation of marine oil snow and thereby played an important role in determining the fate of spilled oil [107,108]. Moreover, this is the first protein-based evidence of a marine HC degrader utilizing alkanes, other HCs and surfactants from Corexit as carbon substrates, while adapting to Corexit exposure by additionally upregulating chemotactic motility, uptake and efflux transporters. These results align with the previously theorized metabolic contributions and chemotactic capabilities of *Marinobacter* members during the DWH oil spill [34,64]. The detected metabolism of Corexit-derived *n*-alkanes also opens up a new perspective on the previously reported stimulation of *n*-alkane biodegradation after chemical dispersant application in some microcosm studies (e.g., in Reference [15]), since observed stimulation effects could be related to the biodegradation of bioavailable *n*-alkanes from the proprietary dispersant mixture as opposed to a presumed more efficient biodegradation of oil components. Finally, our findings suggest that, even though strain TT1 was able to metabolize certain Corexit components, a number of additional cellular adjustments were necessary due to Corexit-induced stress experienced by the cells, such as surfactant/solvent-stress potentially affecting cell membrane functioning. Therefore, our findings help to explain how Corexit can inhibit certain marine HC degraders, as shown in other reports, and how it can enrich for other suspected Corexit-degrading bacteria, thus substantially altering seawater microbial community dynamics during oil spill scenarios when Corexit is used, as documented for various marine habitats [3,4,5,6,7,8]. The short- and long-term ecological consequences of these shifts in composition and functional properties of marine microbial communities require further study and should be taken into consideration for future oil spill response actions.

## Figures and Tables

**Figure 1 microorganisms-09-00003-f001:**
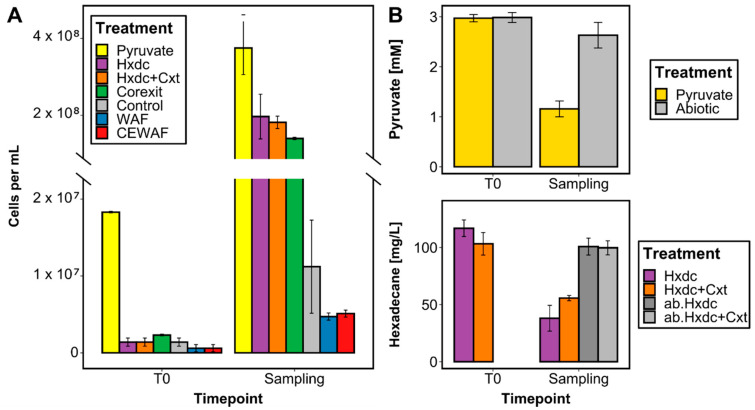
Growth and biodegradation of carbon sources supplied to *Marinobacter* sp. TT1 cultures, sampled for proteomic analysis after 1 day (pyruvate treatments) or 4 days (all other treatments) of incubation. Treatments contained the following carbon sources: 3 mM pyruvate (Pyruvate), 100 mg L^−1^
*n*-hexadecane (Hxdc), 100 mg L^−1^
*n*-hexadecane and 10 mg L^−1^ Corexit (Hxdc+Cxt), 100 mg L^−1^ Corexit (Corexit), no carbon source (Control), 6 mg L^−1^ water-accommodated fraction (WAF)-derived dissolved organic carbon (DOC) (WAF) or 6 mg L^−1^ chemically-enhanced WAF (CEWAF)-derived DOC (CEWAF). Results shown are averages of sacrificial, triplicate cultures (standard deviations are based on triplicates). ab. = abiotic controls without inoculum. (**A**) Cell numbers determined by fluorescence microscopy are presented using a divided *y*-axis in order to better visualize the lower cell numbers in WAF and CEWAF treatments. (**B**) Remaining pyruvate and *n*-hexadecane concentrations were determined via HPLC or GC-MS measurements, respectively.

**Figure 2 microorganisms-09-00003-f002:**
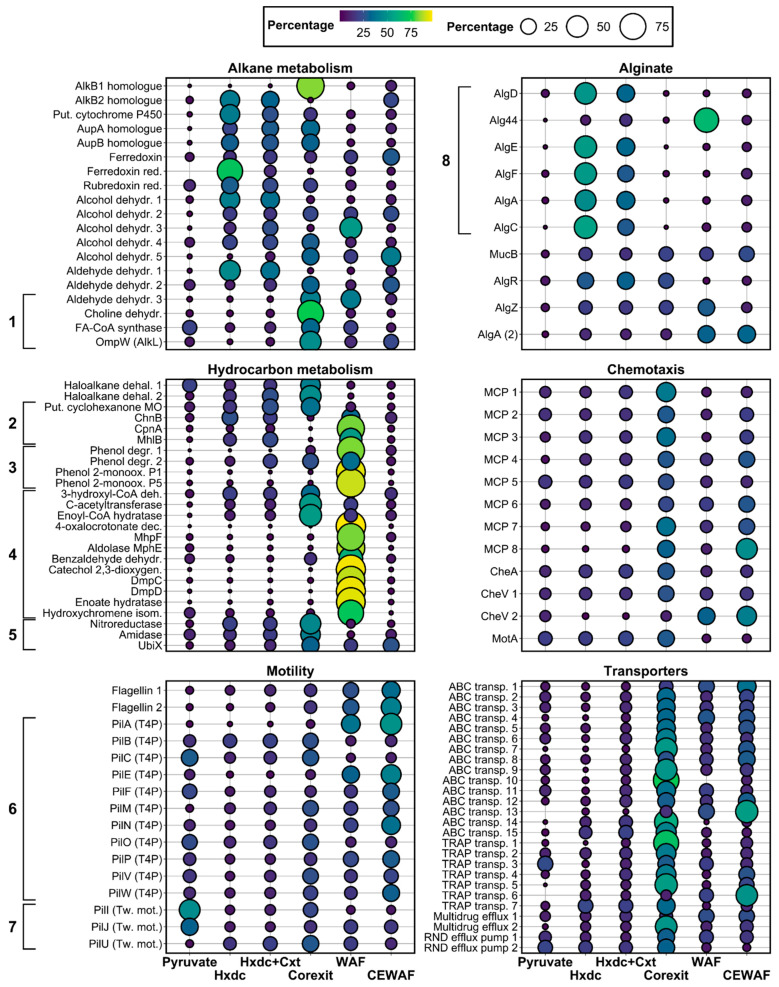
Normalized, relative mean abundances (symbolized by circle size and color; sum per protein = 100%) of a selection of significantly (*q*-value < 0.05) differentially abundant proteins associated with alkane metabolism, alginate synthesis, non-alkane hydrocarbon (HC) metabolism, chemotaxis, motility, and transmembrane transport systems during growth of *Marinobacter* sp. TT1 cultures on different carbon sources. Treatments received either pyruvate, *n*-hexadecane (Hxdc), *n*-hexadecane and Corexit (Hxdc+Cxt), only Corexit (Corexit), crude oil WAF, or chemically enhanced WAF (CEWAF). 1 = proteins belonging to the proposed *alk* operon (plus AlkB1 homologue); 2 = Cycloaliphatic HC metabolism; 3 = Phenol metabolism; 4 = Aromatic HC metabolism; 5 = Aminobenzoate metabolism; 6 = Type IV pilus assembly; 7 = Twitching motility; 8 = Proteins belonging to the proposed *alg* operon. See Appendix A for protein names.

**Figure 3 microorganisms-09-00003-f003:**
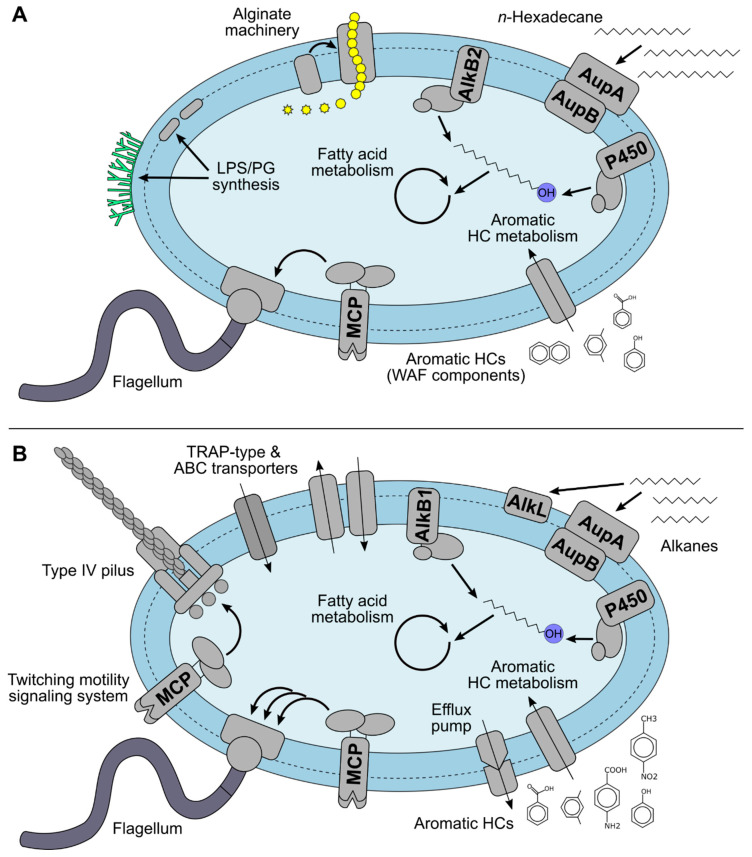
Schematic overview of the metabolism of *Marinobacter* sp. TT1 when (**A**) utilizing hydrocarbons (*n*-hexadecane or WAF), or (**B**) growing on components of Corexit and/or with Corexit exposure. Abbreviations: LPS = lipopolysaccharide, PG = peptidoglycan, HC = hydrocarbon, WAF = water-accommodated fraction, MCP = methyl-accepting chemotaxis protein, TRAP = tripartite ATP-independent periplasmic, ABC = ATP-binding cassette. See Appendix A for protein names.

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
