# Peer review of "Comparative Proteomics of Marinobacter sp. TT1 Reveals Corexit Impacts on Hydrocarbon Metabolism, Chemotactic Motility, and Biofilm Formation"

_microorganisms, 2020, doi:10.3390/microorganisms9010003_

Round 1
Reviewer 1 Report
The manuscript by S. Kleindienst et al. focus on the metabolism of a very interesting microorganism for the biotechnology field and environment. The authors identified for the first time the proteins associated with alkane metabolism and alginate biosynthesis in Marinobacter species, which might be involved in aromatic hydrocarbon biodegradation.
The work present may have considerable impact on the environment and how oil spill is dealt in the future, besides opening avenues for the study and application of the identified pathways.
The manuscript is very well written, and the data presented is sound and well discussed, with the studies and analysis being very complete.
Therefore, I only minor questions to the authors:
- The figure presented in the main manuscript should be of better quality (higher resolution). The one in the supplementary materials seems to be much better.
- Abstract: Sentence starting by “Here, we performed ….. (CEWAF; with Corexit).” Should be made in shorter sentences, as it is quite difficult to understand and the reader is lost.
- Materials and Methods. 2.1. second paragraph , change to “They were adapted to grow on one …”
Reviewer 2 Report
This article describes a study aiming at understanding the mechanisms of the inhibition of alkane biodegradation by dispersants that are used during marine oil spills. A proteomic analysis of Marinobacter sp. TT1 cells grown on pyruvate, hexadecane, or oil in the form of a water-accommodated fraction in the presence or not of the dispersant Corexit was conducted.
This is an interesting work; the rationale of the experiments is logical and sound. Proteomic data are not over-interpreted and confirm several aspects of the physiology of hydrocarbon degradation demonstrated in other studies and more importantly reveal new potential mechanisms of the effect of dispersant on hydrocarbon degradation. The discussion part of this paper is well documented and propose hypotheses that are worth to be tested in further studies. These experiments are of interest in the field of hydrocarbon biodegradation in marine environment. The only general criticism I would have on this study is that it relies exclusively on proteomic data and therefore remains speculative. It would be interesting to have the data on the effects of Corexit on the physiology of Marinobacter sp. TT1, especially on the degradation of hexadecane. The authors claim they have shown that Corexit affects growth and biodegradation of hexadecane by Marinobacter sp. TT1 but they refer unpublished results. Such data would greatly improve the interpretation of the proteomic results and the comprehension of the Corexit effects.
